# Study on Diesel Low-Nitrogen or Nitrogen-Free Combustion Performance in Constant Volume Combustion Vessels and Contributory

**Qinming Tan** [1,*] **, Yihuai Hu** [1] **and Zhiwen Tan** [2]

[1] Merchant Marine College, Shanghai Maritime University, Shanghai 201306, China; yhhu@shmtu.edu.cn
[2] Marine Design & Research Institute of China, Shanghai 201913, China; maric_tanzw@163.com
* Correspondence: qmtan@shmtu.edu.cn; Tel.: +86-021-38282967

**Abstract:** This paper studies the combustion performance of diesel in constant volume combustion vessels under different conditions of mixed low-nitrogen ($O_2$ and $N_2$) or non-nitrogen ($O_2$ and $CO_2$) in varying proportions. The high-speed camera is used to shoot the combustion flame in the constant volume combustion vessel. The process and morphology of the combustion flame are amplified in both time and space to study and analyze the effects of different compositions and concentrations in gases on the combustion performance of diesel and conduct a study on the contributory factors in the performance of diesel with no nitrogen. According to the study, in the condition of low nitrogen, the $O_2$ concentration is more than 60%, the ignition delay period is shortened, the combustion flame is bright and slender, it spreads quickly, and the blue flame appears when the $O_2$ concentration reaches 70%; While for nitrogen-free combustion, only when the $O_2$ concentration reaches 30% is the combustion close to the air condition; when the $O_2$ concentration reaches 40%, the combustion condition is optimized obviously and the combustion flame is relatively slender compared to the air working condition. Similarly, with the increase of the $O_2$ concentration, the ignition delay period of nitrogen-free diesel is shortened, the duration is extended, and the combustion performance is optimized. In addition, when the $O_2$ concentration reaches 50%, with the decrease of the initial temperature, the ignition delay period is prolonged, and the duration is shortened obviously. When the temperature is lower than 700 K, there is no ignition. The increase of the diesel injection pressure is beneficial to optimize the ignition performance of diesel non-nitrogen combustion and shorten its ignition delay period and combustion duration. Related research has important guiding significance to optimize nitrogen-free combustion technology, which produces no NOx of the diesel engine.

**Keywords:** constant volume combustion vessel; diesel engine; nitrogen-free; low-nitrogen; combustion

## 1. Introduction

With the shortage of natural resources and the deterioration of the environment, people's awareness of energy conservation and environmental protection has gradually increased [1]. Diesel engines, as one of the main discharge equipment for pollutants, have drawn people's attention [2,3]. Many laws and regulations at home and abroad have put strict requirements on the carbide and nitrogen oxide emissions, and effectively promote the research and development of energy conservation and emission reduction technologies of diesel engines [4–6]. At present, there are technologies based on intake components, such as EGR [7,8], intake oxygen enrichment/hypoxia [9–12], intake air humidification [13,14], and in-cylinder water injection [15,16] to optimize the combustion and emission performance of diesel engines. All these technologies are based on the change of gas components in the combustion environment. To reduce NOx, the temperature in the combustor should be changed in the combustion process at the expense of the economics of the diesel engine [17,18]. Some research institutes have carried out comprehensive optimization of EGR and oxygen-enriched air intake, which can make SOOT-NOx emission

lower than the original machine; however, the result is not obvious [19,20]. Therefore, the SOOT-NOx emission bottleneck has become the key to restrict the development of combustion and emission technologies of diesel engines [21,22]. The $O_2$ and $N_2$ concentration in the combustion environment is of great significance for the production of NOx [23,24]. This paper aims to find new solutions to optimize diesel emissions through basic studies on diesel low-nitrogen or nitrogen-free combustion.

Since Home and Steinburg first proposed the nitrogen-free coal combustion technology in 1981, relevant applications and research has been carried out at home and abroad [25,26]. Its NOx emissions are much smaller than air conditions, which has crucial reference value for nitrogen-free combustion studies in other fields [27,28]. At present, there are relatively few studies on nitrogen-free combustion of the engine. Wang Zuofeng and other researchers carried out nitrogen-free combustion experiments and numerical modelling studies based on the diesel engine ZS195, a single-cylinder machine. The result showed that the diesel engine can idle and reduce NOx emission (approximately zero) under the inlet condition of 60% $O_2$ and 40% $CO_2$. Its HC and CO emissions just slightly increased compared to air conditions. When the $O_2$ concentration is lower than 50%, the diesel engine cannot be on fire, and when the $O_2$ concentration reaches 65%, the inlet condition becomes relatively better, and can be optimized by the change of the fuel supply advance angle [29,30]. Zhao Tianpeng and other researchers conducted a numerical simulation calculation of diesel and a visualization study of the constant volume combustion vessel, which showed that in the condition of 65% $O_2$ and 35% $CO_2$, the ignition delay period of diesel was reduced by 50% compared to air conditions. In the condition of 50% $O_2$ and 50% $CO_2$, diesel can also be on fire and the ignition delay period is shortened by 35% [31,32]. Tan Qinming and others carried out numerical simulation calculations and bench tests under a low-nitrogen and nitrogen-free environment based on the 4135ACa diesel engine. The result showed that the diesel engine can start and operate continuously under the condition of 50% $O_2$ and 50% $CO_2$, but the combustion was abnormally bad and the fuel consumption rate was significantly higher compared to air conditions. When the $O_2$ concentration increased to 60%, the combustion improved but was not perfect, and even visible carbides appeared [33,34]. Zhu Changji and other people carried out experimental studies on the characteristics of homogeneous premixed combustion of gasoline under a nitrogen-free environment, which showed that the ignition delay period and duration were shortened by 50%, and the pressure growth rate increased by 180% under the condition of 40% $O_2$ and 60% $CO_2$ [35,36]. The above research results of nitrogen-free combustion are basically consistent. Different test conditions led to different results [29,36]. At present, there are few studies on the factors that affect diesel nitrogen-free combustion.

Experiments were performed in a constant volume combustion vessel, with fuel injected into the vessel at selected mixed air, pressure, and temperature conditions [37,38]. In this paper, the low-nitrogen or nitrogen-free mixed gases in varying proportions are used to inflate the fixed volume combustion vessel, and the mixed gases are heated to a certain temperature so that the fuel can be ignited and burned. The high-speed camera is used to shoot the combustion flame in the constant volume combustion vessel. The combustion flame is analyzed from time and space. The performance of diesel that has low nitrogen and is nitrogen free is discussed from the aspects of flame morphology, ignition delay period, combustion duration, and flame intensity [39–41]. In addition, this paper studies the effects of the initial temperature of the mixer and the fuel injection pressure on the diesel nitrogen-free combustion performance, which provides an important guide to optimize the technology that produces no NOx of diesel engines.

## 2. Experimental and Test Method

### 2.1. Experimental Method

Many experimental works have been performed using constant volume combustion bombs with adjustable initial conditions [42–44]. The principle of the constant volume combustion vessel and its visual test system are shown in Figure 1. The maximum volume

of the constant volume combustion vessel is 43 L. The effective volume is 12 L and the maximum pressure is 60 bar. Various fuel injection and combustion tests can be performed. Different proportions of mixed gases are used to inflate the bomb, and heat it through the heating tile. The maximum heating temperature can reach 900 K, and the accuracy is $\pm 10$ K. The constant volume system is equipped with a 0.14 mm single injector orifice. The fuel high-pressure common rail injection system provides an injection pressure of 600-1800 bar and can regulate the injection cycle, times, pressure, and pulse. The Schott HAB-150W halogen lamp was used as the constant light source during the test. The Photron FASTCAM SA5 CMOS high-speed camera was used to collect images of the flame in the constant volume combustion vessel during the combustion process under different conditions. The shooting speed was 10,000 fps. The image resolution was $768 \times 768$ and the low f number was 17. The fuel injection of the entire experimental bench, high-speed camera shooting, and the collection of thermal parameters were synchronously triggered to ensure the synchronization of data acquisition under different working conditions.

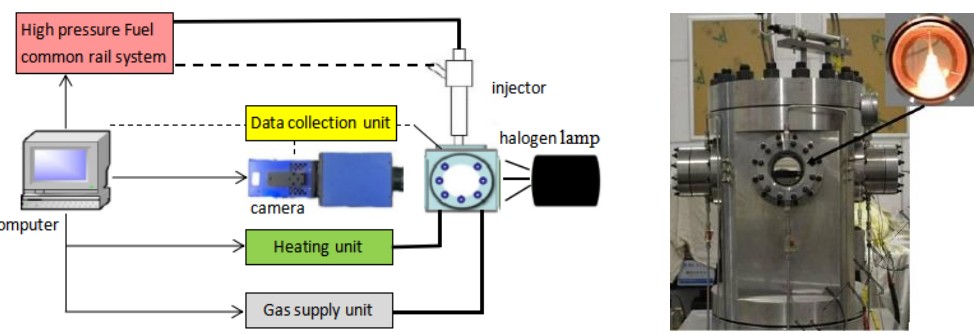

**Figure 1.** The visual test system of the constant volume combustion vessel.

### 2.2. Test Method

The low-nitrogen oxygen-enriched tests of the constant volume combustion vessel were carried out mainly under conditions of 60% $O_2$ and 40% $N_2$, 70% $O_2$ and 30% $N_2$, and 80% $O_2$ and $N_2$ 20% to explore the limit of the oxygen enrichment concentration in low-nitrogen combustion that can reduce NOx emission under the premise of combustion safety. For nitrogen-free combustion, tests were carried out by mixing gases of 21% $O_2$ and 79% $CO_2$, 30% $O_2$ and 70% $CO_2$, 40% $O_2$ and 60% $CO_2$, and 50% $O_2$ and 50% $CO_2$. The initial pressure of mixed gases in the bomb was 40 bar, the initial temperature was 800 K, and the diesel was injected with 16 mg with 1200 bar pressure to conduct the combustion test. Under the nitrogen-free environment of 50% $O_2$ and 50% $CO_2$, two experiments were studied in order to optimize the nitrogen-free combustion performance of the diesel engine: firstly, the diesel nitrogen-free combustion performance (with 1200 bar injection pressure) of diesel fuel with the gas mixture temperature of 750 or 700 K, and secondly, the effect of 900 bar or 600 bar diesel injection pressure on its nitrogen-free combustion performance (with the temperature of 800 K of gas mixture). Through the process and analysis of the diesel combustion flame images taken by the high-speed camera, the effects of the $O_2/N_2/CO_2$ concentration, initial mixture temperature, and diesel injection pressure on the diesel combustion process were studied.

Pictures taken by the high-speed camera were processed as follows: take a screenshot and draw a scale to analyze the length, width, and appearance of the burning flame; convert the picture to a grayscale picture; and set pictures less than 5 to 0 intensity to eliminate the influence of the background value. By calculating the total light intensity value of each horizontal row, the starting moment of the combustion flame and the combustion duration with total light intensity exceeding 100 cd were analyzed. Studying the change law of the combustion start position of the flames at any time allowed comparative analysis of the total light intensity of the flame.

In order to ensure the validity of the data of the diesel engine's ignition delay period, the high-speed camera was triggered at the same time of the fuel injection to perform

synchronous shooting. Pictures were taken at a certain frequency, and the data were averaged by multiple injection combustion tests. Due to the large volume of gas working fluid in the bomb (40 L), the release heat of 16 mg diesel combustion showed too little influence on the gas working fluid pressure in the constant volume combustion vessel, and the relevant exothermic analysis was not performed. Additionally, the total volume of the waste gas after the single combustion test was not enough for the detection and analysis of NOx, carbide, and other emissions. Relevant experiments can be further carried out under favorable experimental conditions.

## 3. Results and Discussion

### 3.1. Low-Nitrogen Combustion

Under low-nitrogen ($O_2$ and $N_2$) conditions, part of the photos of flame during the diesel combustion process is shown in Figure 2. The first photo of the air condition was taken at 1.6ms after the injection, and the other conditions were taken 0.6ms after the injection. The time interval was 0.2 ms. Compared with the flame in air condition, when the $O_2$ concentration reached 60%, the ignition delay period was shortened from 2.07 ms to 1.06 ms, the flame brightness was obviously enhanced, the flame shape was slender, the longest flame length was about 60 mm, and the width was about 10 mm. The flame shape of the air working condition was relatively large, and the longest flame was about 50 mm long and the maximum width was nearly 40 mm. It showed that in the condition of high concentration of oxygen enrichment, the combustion flame is significantly faster than the air condition. Especially when the $O_2$ concentration reached 70% or 80%, the burning moment was further advanced, and a blue flame appeared. However, there was no abnormal phenomenon in the constant volume combustion vessel body. In this test, the amount of air was large, and the amount of fuel injection was small, the heat of combustion was limited, and the temperature of the combustion flame could not reach the temperature of the body of the damaged constant volume. The above results can effectively support the phenomenon mentioned in the literature [33]: the sparking phenomenon of the exhaust pipe in the diesel engine appeared under the 60% $O_2$ oxygen-enriched and low-nitrogen conditions.

In order to quantitatively analyze the combustion process, data of the total light intensity, combustion start position (the distance between the first burning flame and the injector nozzle), ignition delay period, and duration are shown in Figure 3 and Table 1 by processing the photos. The increase in total light intensity under low-nitrogen conditions advances, and the steep curve indicates that the rate of rise is much faster than the air condition. The maximum light intensity under low-nitrogen conditions is slightly larger than that of air, and the maximum intensity of light intensity lasts longer than 2 ms. The flat-top curve is strikingly different from the singe-peak curve of the air working condition. The reason is that the concentration of $O_2$ in the combustion environment rises greatly, which is beneficial to the ignition and combustion of diesel fuel. As a result, the combustion propagation speed accelerates, and the efficiency improves. The high-efficiency combustion in the early stage of low-nitrogen combustion accelerated the consumption of diesel fuel, resulting in a decrease in the amount of diesel in the late combustion stage, which causes the sharp fall in total light intensity, rather than a slow decline in the air condition. It is worth noting that the total intensity of light at the same time as low-nitrogen combustion conditions does not increase with the rise of the $O_2$ concentration. The total intensity of light under 70% or 80% $O_2$ concentrations are lower than 60%. The main reason for this phenomenon is that when the $O_2$ concentration reaches a certain value, the combustion condition of diesel reaches the limit. This phenomenon is caused by thermal-physical differences between $O_2$ and $N_2$.

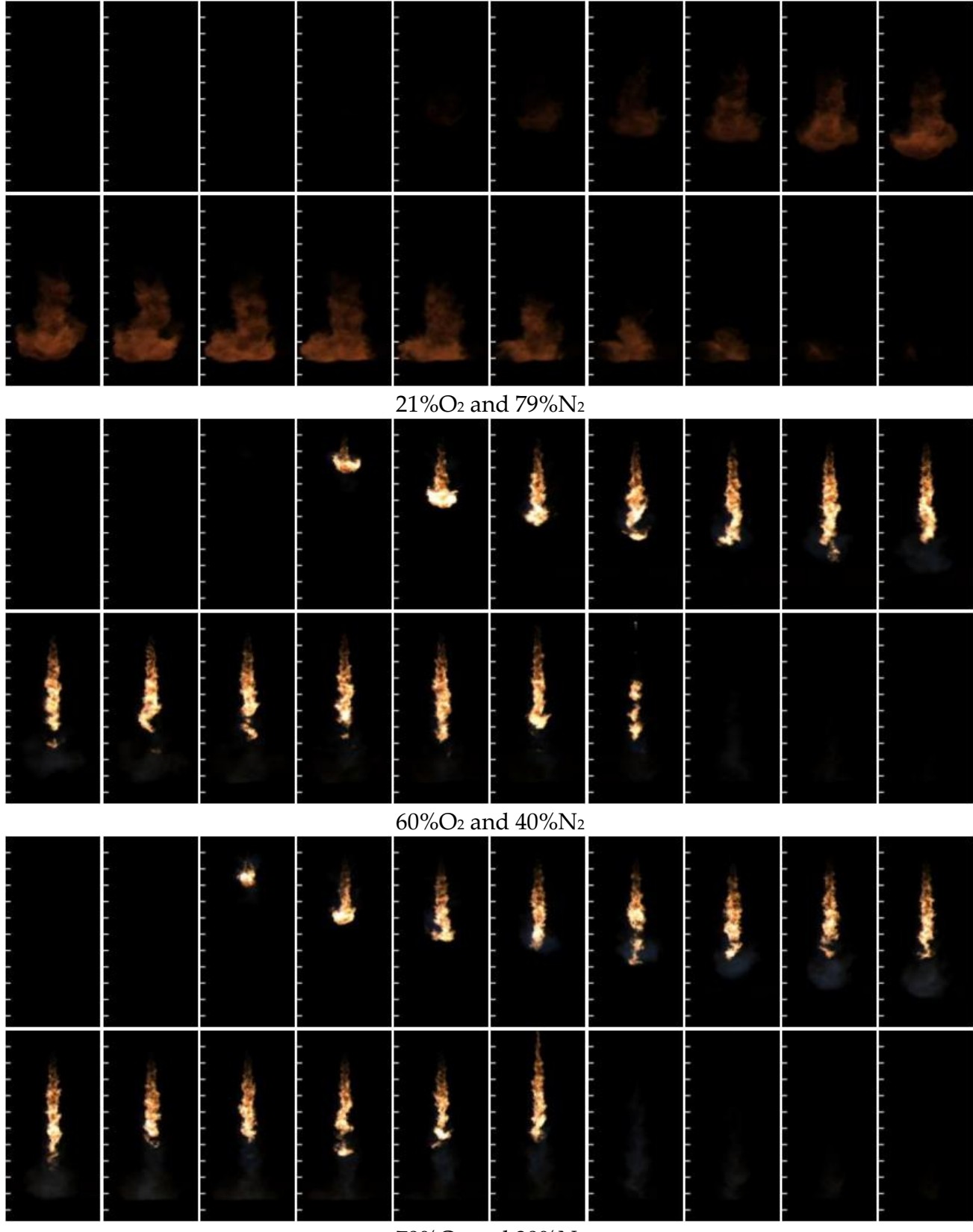

21%$O_2$ and 79%$N_2$

60%$O_2$ and 40%$N_2$

70%$O_2$ and 30%$N_2$

**Figure 2.** Comparison of combustion flames under different low-nitrogen conditions.

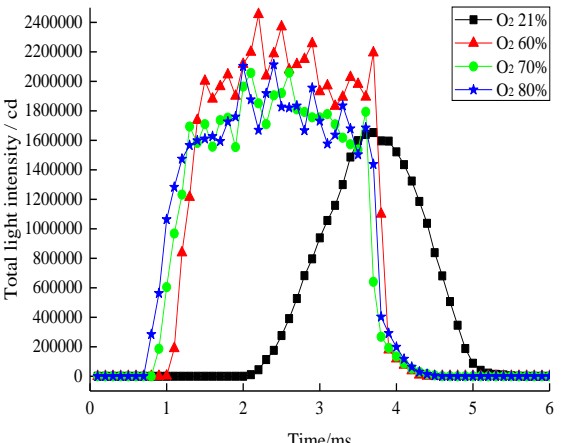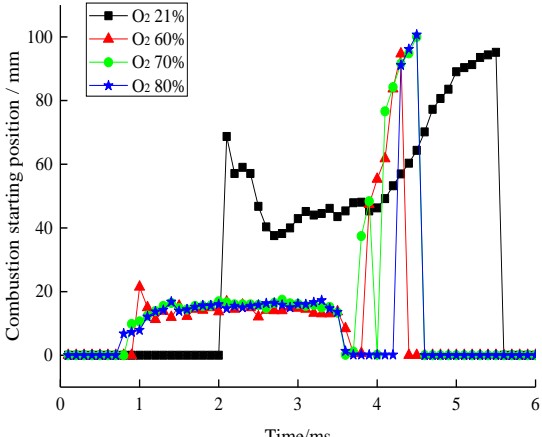

**Figure 3.** Comparison of the total light intensity and combustion starting position of combustion.

**Table 1.** Comparison of the ignition delay period and duration of constant volume combustion vessels under low-nitrogen conditions.

| Parameter Types | 21% $O_2$ and 79% $N_2$ | 60% $O_2$ and 40% $N_2$ | 70% $O_2$ and 30% $N_2$ | 80% $O_2$ and 20% $N_2$ |
|---|---|---|---|---|
| Ignition delay period/ms | 2.07 | 1.06 | 1.03 | 0.98 |
| Standard deviation of ignition delay | 1.57 | 1.17 | 1.7 | 1.75 |
| Combustion duration/ms | 3.3 | 3.4 | 3.5 | 3.6 |
| Standard deviation of combustion duration | 1.45 | 1.39 | 0.5 | 0.83 |

The decrease in the ignition delay period is sure to make the starting position is closer to the injector nozzle. In case the $O_2$ concentration is 60%, the distance between the flame and the injector nozzle is less than 20 mm and it will be further shortened with the further increase of the $O_2$ concentration, which is consistent with the change law of the ignition delay period. Additionally, the starting position of the flame stays close to the injector nozzle, and does not move down with the combustion process until the flame is extinguished in the later stage of combustion. The position of the combustion flame in the air condition gradually moves down as the combustion process progresses, which is especially fast in the later stage of combustion. Therefore, it is further explained that the combustion speed and efficiency of the diesel in the low-nitrogen high-concentration oxygen-enriched conditions are significantly better than the air conditions.

The ignition delay period and combustion duration of four combustion processes under the same gas working condition were averaged and analyzed as shown in Table 1. The standard deviation of the ignition delay period of each working condition is about 1.5 ms, and the standard deviation of the burning duration is about 1, indicating that the data are relatively stable. The average ignition delay period of air conditions is about 2.07 ms, and when the $O_2$ concentration reaches 60%, the ignition delay period is obviously shortened to 1.06 ms. With the increase of the $O_2$ concentration, the period is further shortened, and when the $O_2$ concentration reaches 80%, the burning period is only 0.98 ms.

### 3.2. Nitrogen-Free Combustion

The constant volume combustion vessel test was carried out by replacing the $N_2$ (21% $O_2$ and 79% $CO_2$) in the air with $CO_2$. As shown in Figure 4, the first photo of each working condition was taken at 1.6 ms after the injection. Compared with Figure 3, although the combustion time is not much different from the air condition (about 2.1 ms), the combustion flame was very dark and the flame shape was slightly smaller. It indicates that the presence of a great deal of $CO_2$ inhibits the combustion chemical reaction that produces $CO_2$, thereby inhibiting the combustion process. Additionally, the $CO_2$ heat transfer performance is

much lower than that of $N_2$, which weakens the flame propagation. This test indicates that diesel undergoes a chemical reaction in a 21% $O_2$ and 79% $CO_2$ environment.

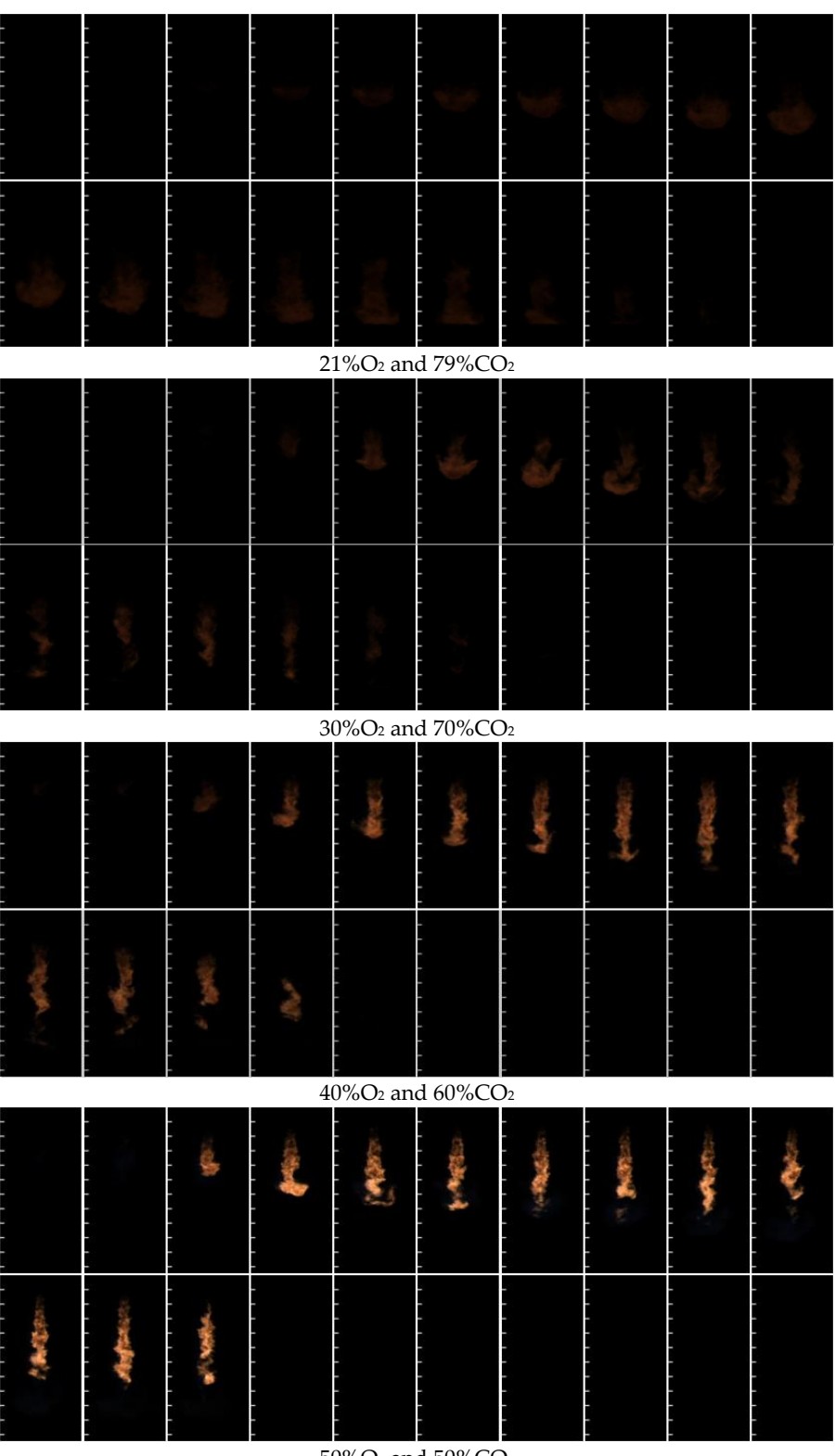

21%$O_2$ and 79%$CO_2$

30%$O_2$ and 70%$CO_2$

40%$O_2$ and 60%$CO_2$

50%$O_2$ and 50%$CO_2$

**Figure 4.** Comparison of combustion flames under different nitrogen-free conditions.

The constant volume combustion vessel test was carried out by increasing the $O_2$ concentration to 30%, 40%, or 50% in a nitrogen-free environment, as shown in Figure 4.

With the increase of the $O_2$ concentration, the diesel-free ignition delay period was short-ened by nearly 0.6 ms, and the combustion flame gradually became brighter. When the $O_2$ concentration reached 30%, the combustion flame became obviously better, but it was still inferior to the air working condition, and was especially weak and discontinuous in the late combustion stage. When the oxygen concentration reached 40%, the combustion flame brightness was close to the air working condition. Additionally, its flame appearance became slender. It shows that the flame propagation speed increases as the $O_2$ concentra-tion rises. In particular, the flame in the 50% $O_2$ condition is slender and bright, indicating that the nitrogen-free combustion in the 50% $O_2$ concentration condition can continue and the combustion is perfect. The ignition delay period and combustion duration of the five combustion processes in each gas working condition were averaged and analyzed, as shown in Table 2. The standard deviation between the ignition delay period and the combustion duration of each working condition is about 1.5 ms, indicating that the data are stable. In working conditions of 21%, 30%, 40%, and 50% $O_2$, the non-nitrogen ignition delay period is 2.01, 1.59, 1.46, and 1.18 ms, respectively. Therefore, the ignition delay period is mainly affected by the $O_2$ concentration in the environment. The combustion duration was shortened from 3.2 to 2.7 ms.

As shown in Figure 5, during the diesel combustion process of 21% $O_2$ and 79% $CO_2$, the total intensity of light at each moment did not exceed 20,000 cd; there was a big difference from air condition. When the $O_2$ concentration of nitrogen-free combustion increased to 30%, the combustion start time was earlier than the air condition, that is, the ignition delay period was shortened from 2.07 to 1.59 ms. Additionally, the total light intensity rise curve was parallel to the air condition rise curve, but the total intensity of light at each moment was lower than that of the air condition. When the total intensity of the light gradually decreased after reaching its maximum value, it was consistent with the air condition, mainly due to the small amount of diesel combustion at an early stage. Therefore, the late combustion continued. When the $O_2$ concentration increased to 40%, the total light intensity rise rate was more than the air condition, and the total light intensity continued to maintain a relatively high position, and then rapidly decreased. It indicates that the diesel combustion under this working condition is mainly at the early and middle stage, but its maximum light intensity is still significantly lower than the air condition. When the oxygen concentration reaches 50%, the combustion advances and the total intensity of the light rises rapidly to a high point (close to air conditions), and the maximum intensity of light intensity lasts for about 1.8 ms, which is longer than the single-peak curve of the air conditions. The total intensity of the combustion flame indirectly reflects the diesel combustion, which indicates that the diesel combustion is better than the air condition under this working condition.

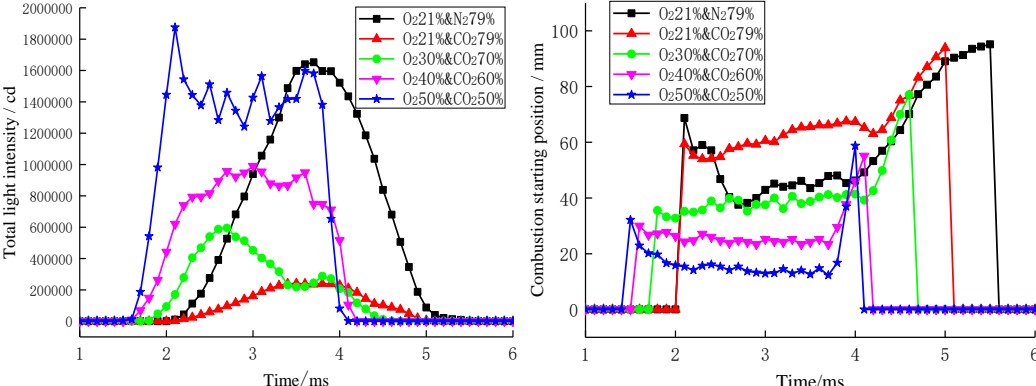

**Figure 5.** Comparison of the total light intensity and combustion starting position of combustion flames at various times in nitrogen-free conditions.

**Table 2.** Comparison of the ignition delay period and duration of constant volume combustion vessels under nitrogen-free conditions.

| Parameter Types | 21% $O_2$ and 79% $N_2$ | 21% $O_2$ and 79% $CO_2$ | 30% $O_2$ and 70% $CO_2$ | 40% $O_2$ and 60% $CO_2$ | 50% $O_2$ and 50% $CO_2$ |
|---|---|---|---|---|---|
| Ignition delay period/ms | 2.07 | 2.01 | 1.59 | 1.46 | 1.18 |
| Standard deviation of ignition delay | 1.57 | 1.38 | 1.66 | 1.65 | 0.41 |
| Combustion duration/ms | 3.3 | 3.2 | 2.9 | 2.7 | 2.7 |
| Standard deviation of combustion duration | 1.45 | 2.57 | 0.25 | 1.49 | 1.56 |

Although the diesel combustion time of 21% $O_2$ and 79% $CO_2$ was close to the air condition, the distance between the combustion flame and the injector nozzle was obviously larger than the air condition. When the O2 concentration increased to 30%, the start position of the combustion flame was similar to the air condition. When the $O_2$ concentration was less than 30% at the middle stage of the nitrogen-free combustion, the combustion flame gradually moved down as the combustion processed, which was consistent with the air condition, as shown in Figure 5. The $O_2$ concentration of the 40% and 50% nitrogen-free combustion flame start position will stay at a certain position. Especially, the nitrogen-free combustion condition of the 50% working condition is close to the low-nitrogen combustion condition. The combustion flame starts to stay at the position that is less than 20 mm from the injector nozzle.

### 3.3. Factors Affecting Nitrogen-Free Combustion Performance

The above research shows that the diesel combustion process is superior to air conditions when diesel is injected at a fuel injection pressure of 1200 bar in a 50% $O_2$ and 50% $CO_2$ nitrogen-free environment at 800 K. The current relevant research results show that the diesel combustion in the 50% nitrogen-free environment is worse while when the oxygen concentration reaches 65%, the combustion condition is better. Even the literature 29–31 shows that the combustion condition of diesel engines under the 60% intake condition is not perfect, and there are even visible carbon particles. The main reason for the inconsistency of the above conclusions is the influence of the initial temperature of the mixture at the time of injection and the diesel injection pressure. The following studies will be carried out to provide a reference for the optimization of the diesel-free combustion performance of diesel at the later stage.

#### 3.3.1. Initial Temperature

Parts of the photos of the combustion flame taken at a temperature of 800 or 750 K in a constant volume combustion vessel are shown in Figure 6. The first photo is the burning photo corresponding to 1.6 ms after the injection, and the interval is 0.2 ms. It can be seen from the figure that the diesel ignition delay period in the condition of 750 K is quite long (about 2.9 ms), which is much longer than 1.9 ms under the condition of 800 K. Additionally, the burning duration was very short, only about 1.2 ms, while the intensity of the flame was very weak, and the burning flame was not captured by the aperture of 17. In Figure 6, the flame at 750 K was shot by an aperture of 11. The total intensity of the light was approximately twice of that of the 17-aperture photograph. From Figure 6, visually, the intensity of the flame's light shoot by 17 aperture in the condition of 800 K was a little difference from that in the condition of 750 K. When the temperature lowered to 700 K, the image of the combustion flame inside the constant volume combustion vessel was not captured by any aperture. It indicates the initial temperature of the nitrogen-free mixed gases is critical to the combustion performance of diesel. The specific heat capacity of $CO_2$ is relatively large. As a result, a great deal of $CO_2$ in the diesel engine without nitrogen can cause the compression end temperature to be relatively low, thus affecting the nitrogen-free combustion performance of the diesel engine. This is also one of the key factors leading to poor combustion of diesel engines in 50% $O_2$ and 50% $CO_2$ nitrogen-free conditions in the literature 2934. Under the premise of sufficient $O_2$ in the nitrogen-free combustion

environment, the high-temperature smoke and gases emitted by the diesel engine can be recycled by EGR to increase the intake air temperature, and handle the problem of the $O_2$ and $CO_2$ consumption, thereby effectively reducing the operating cost of the diesel engine.

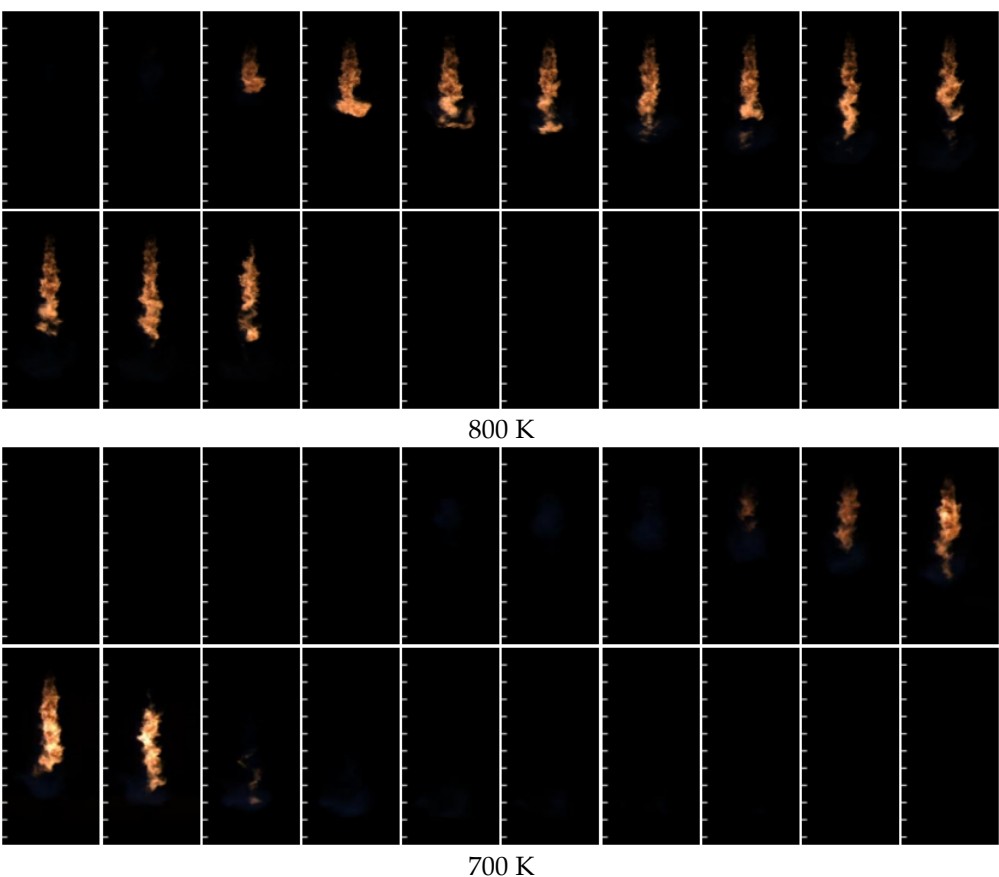

**Figure 6.** Influences of different initial temperatures on the diesel combustion flame in a 50% $O_2$ nitrogen-free environment.

3.3.2. Injection Pressure

A large number of studies have shown that the injection pressure directly affects the progress of combustion [34,40]. This is true of the conclusion drawn from the optimization experiment of a constant volume combustion vessel without nitrogen combustion. As shown in Figure 7, in the nitrogen-free environment of 50% $O_2$ and 50% $CO_2$, the diesel fuel was injected at a pressure of 1200, 900, and 600 bar, respectively, and photos were taken of the combustion flame in a constant volume combustion vessel. The first picture of each working condition was the 16th shot after the injection, and the later interval was 0.2 ms. From the perspective of flame topography, there is not much difference. As the injection pressure decreased, the combustion time was obviously shifted back (1.5, 1.7, 1.9 and ms, respectively), and the combustion duration was extended (corresponding to 2.7, 3.1, and 4.0 ms, respectively). For the total intensity of the light and the starting position of the flame at different times, the analysis of the photograph was obtained, as shown in Figure 8. The change of the injection pressure does not affect the curve shape of the flat-top total intensity, and during the earlier and later period, the rise or drop rates of the total intensity of the light are similar. With the decrease of injection pressure, the maximum total light intensity was higher, and the change was not obvious. However, the increase of the injection pressure led to advancement of the diesel free-nitrogen combustion time, which can significantly improve the ignition performance. The authors of [34] used a Dongfeng 4135 ACa diesel engine, and its injection pressure was only about 300 bar, which is one of the key factors that led to poor combustion of the diesel engine under the nitrogen-free

working condition of 50% $O_2$ and 50% $CO_2$. The above information implies that the diesel injection pressure is also critical for the diesel engine's nitrogen-free combustion ignition performance. It is recommended to use a higher diesel injection pressure in later studies related to nitrogen-free combustion to optimize its ignition and combustion performance.

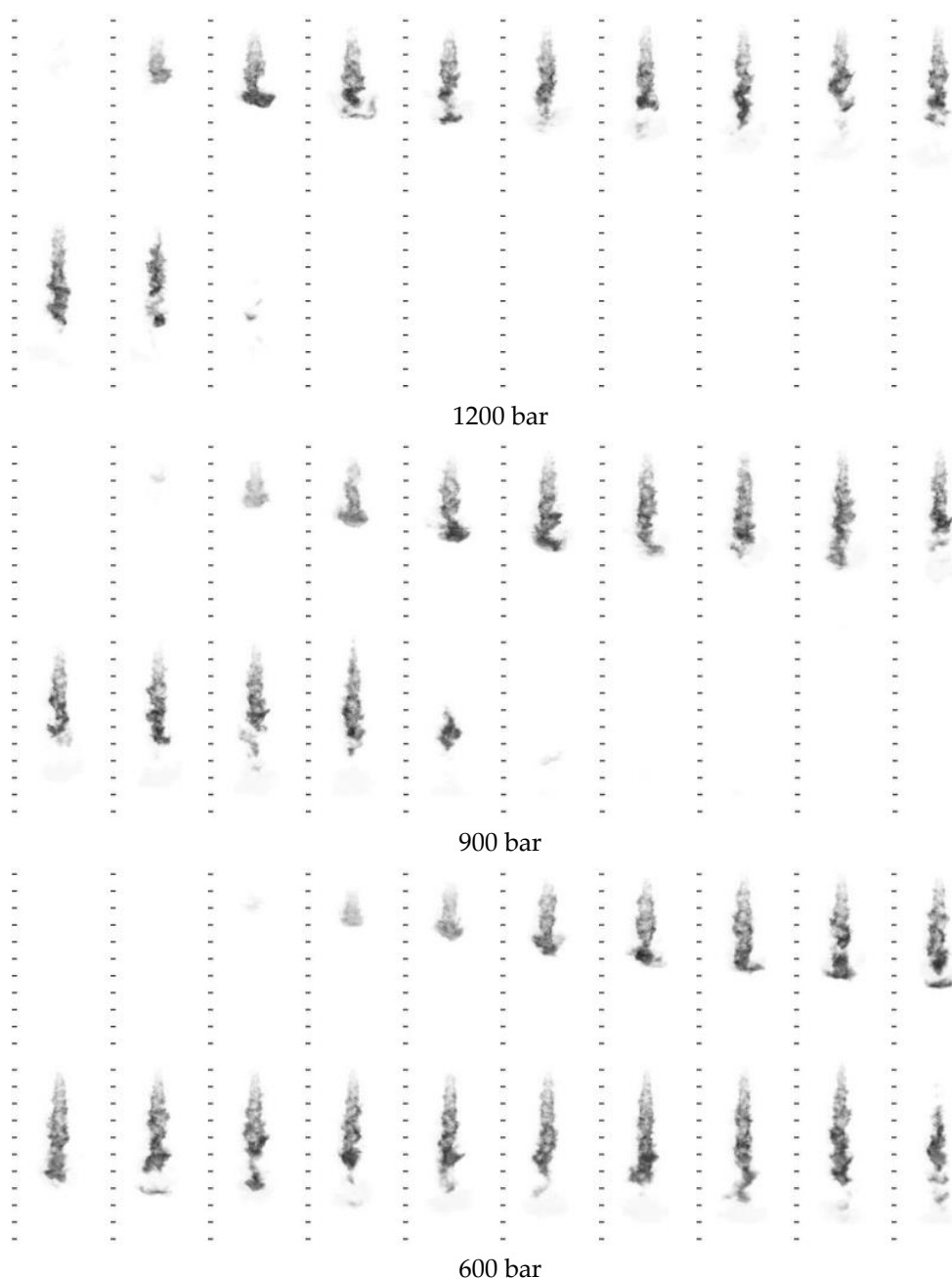

1200 bar

900 bar

600 bar

**Figure 7.** Effects of injection pressure on diesel combustion flame in a 50% $O_2$ nitrogen-free environment.

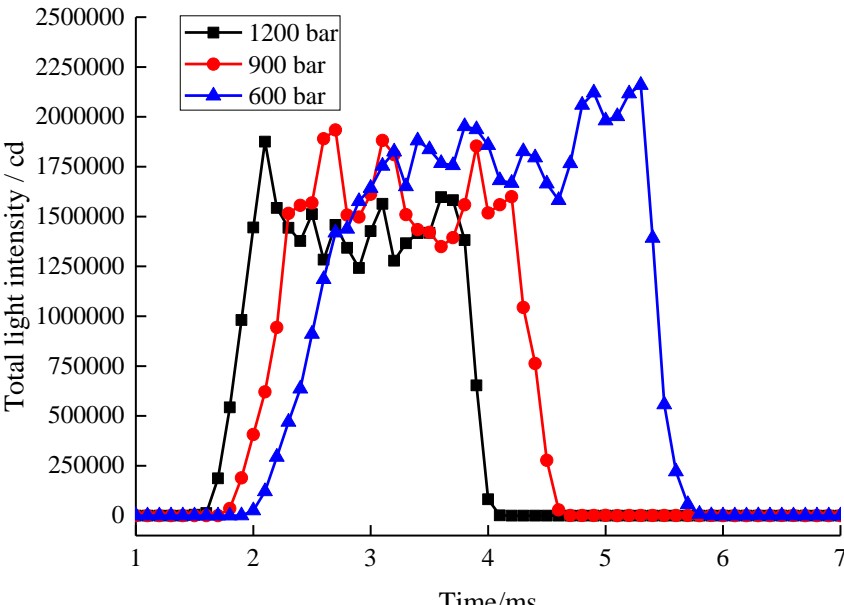

**Figure 8.** Comparison of total light intensity of constant volume combustion vessel in a 50% $O_2$ nitrogen-free environment.

## 4. Conclusions

In this study, low-nitrogen and nitrogen-free combustion tests of diesel constant volume combustion vessels were carried out. The initial pressure in the bomb was 40 bar, the initial temperature was 800 K, and 16 mg of fuel was injected at a pressure of 1200 bar for the combustion test. The following conclusions can be made after processing and analyzing the photos of the diesel combustion flame that were taken by a high-speed camera:

(1) Diesel can be on fire in a low-nitrogen environment with 60% $O_2$ concentration. The flame is bright and slender, in contrast to the dim and large flames of the air condition, and there is no abnormal phenomenon in the combustion. When the $O_2$ concentration reaches 70% or 80%, there is blue flame. Although there is no effect on the body of the constant volume combustion vessel, there may be an abnormality in the high-load condition of the diesel engine. The diesel engine is not suitable for operation in a low-nitrogen environment with too high an $O_2$ concentration.

(2) Under the condition of 21% $O_2$ and 79% $CO_2$, the diesel can be on fire. However, the flame is extremely dim, the combustion duration is short, and the combustion process obviously deteriorates.

(3) The increase of the $O_2$ concentration can optimize diesel nitrogen-free combustion. When the $O_2$ concentration reaches 30%, the combustion time is prior to the air condition, and the increase rate of the total intensity of the flame is similar to the air condition, but the maximum total intensity is still far below air conditions. When the $O_2$ concentration reaches 50%, the combustion flame is bright and slender, and the maximum total intensity of the light is close to the air condition. The maximum total intensity is about 1.8 ms. It shows that under this working condition, diesel can burn normally.

(4) In the nitrogen-free working condition of 50% $O_2$ concentration, diesel burns better at an initial temperature of 800 K, while combustion deteriorates severely at an initial temperature of 750 K and there is no burning phenomenon at the initial temperature of 700 K. The temperature is of great significance for the nitrogen-free combustion of diesel. The decrease in the end temperature of the diesel nitrogen-free intake compression may be the main cause of the abnormal combustion in the 50% $O_2$ working condition.

In the nitrogen-free condition of the 50% $O_2$ concentration, diesel is injected at a pressure of 1200, 900, and 600 bar, respectively, and the combustion time corresponds to 1.5, 1.7, and 1.9 ms, respectively. In addition, the combustion duration corresponds to 2.7, 3.1, and 4.0 ms, respectively. It indicates that the injection pressure is critical to the ignition

performance and duration performance of nitrogen-free combustion. The 300-bar injection pressure of the diesel engine is likely to be the main reason for the insufficient 50% $O_2$ nitrogen-free combustion.

**Author Contributions:** Conceptualization, Q.T. and Y.H.; methodology, Q.T.; software, Z.T.; validation, Q.T., Y.H. and Z.T.; formal analysis, Q.T.; investigation, Z.T.; resources, Q.T.; data curation, Q.T.; writing—original draft preparation, Q.T.; writing—review and editing, Q.T.; visualization, Q.T.; supervision, Q.T.; project administration, Q.T.; funding acquisition, Y.H. All authors have read and agreed to the published version of the manuscript.

**Funding:** This research was funded by the Science & Technology Commission of Shanghai Municipality and Shanghai Engineering Research Center of Ship Intelligent Maintenance and Energy Efficiency, grant number 20DZ2252300.

**Institutional Review Board Statement:** Not applicable.

**Informed Consent Statement:** Not applicable.

**Data Availability Statement:** The study did not report any data.

**Acknowledgments:** Thanks to Liu Haifeng of Tianjin University for his technical support.

**Conflicts of Interest:** The authors declare no conflict of interest. The funders had no role in the design of the study; in the collection, analyses, or interpretation of data; in the writing of the manuscript, or in the decision to publish the results.

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
