# Peer review of "Study on Diesel Low-Nitrogen or Nitrogen-Free Combustion Performance in Constant Volume Combustion Vessels and Contributory"

_atmosphere, doi:10.3390/atmos12070923_

Round 1
Reviewer 1 Report
In my opinion, the explanation of the research presented within this paper is not clear enough. The introduction is not very exhaustive and specific, there is a scarce state of the art, although most of the results are coherent with previous knowledge, I consider that several issues need additional information, the objective of the work is not clear, and some justifications should be added to improve the quality of the manuscript. It does not imply an outstanding innovation in this research field. In this sense you should try to highlight what is the innovation of you approach, and focus the analysis on novelties. Regarding the approach of this study, I consider that it should be very interesting an explanation of what possible implications this research would have, the for example in the automotive industry. Is this research part of a bigger study? Is a long term aim to employing these gases?
- You must carefully review the indicated gas combinations, different errors have been detected in the text, Line 121 40% O2 (it must be N2). In the same way, Figure 5, there is a mistake in the legend, Black line O2 and O2, please check and correct.
- It should be very interesting to translate the camera results to temperature (thermographic camera, for instance).
- Table 1 values. It is not explained how exactly has been quantified all these parameter, from the images collected, the parameters that are analysed, for example, how the start and end of combustion are determined, what is the specific criterion.
- There are some expressions that should be improved in a more quantitative way, for example:
- Line 284, and 289: ‘much lower than’
- Line 318: ‘relatively poor’
- Line 319: ‘relatively good’
- Line 336: ‘was not much different from that`
- Line 371: ’the maximum total light intensity was relatively high’
- Line 379: ‘relatively high diesel injection pressure’
- Line 101. [36-44]. it is not a good practice, using so many references together (8), without specifying anything about them. Please extend these references or reduce the number of cited research.
- Line 167: ‘and a similar blue flame appeared’ It is not a very scientific expression, nothing quantitative is shown, it seems that it is totally conditioned to the observed, the detection system should be improved and use a reliable system.
- Line 176. 'combustion start position', this parameter must be explained, and should be identify in some figure or scheme. In Line 313, you wrote ‘from the injector nozzle’, if this is the distance, it must be specified.
- Line 190: ‘ 70% and 80% O2 concentrations are slightly lower than 60%.’ This sentence does not have sense, I think that the original meaning has been modified, please review.
- Figure 2 and 4. I consider that any text or legend indicating the time of each image would help to understand it. In addition, a greater separation between the images corresponding to different test conditions must be done. Please modify your figures in this way.
- Line 358. A new reference should be included.
- Line 396. ‘in this chapter’. The word ‘chapter’ I think is not adequate.
Minor errors, please review:
- Line 11. an space here is needed. Line 172, an extra space must be included here, please check space again in Line 157, Line 262, 268, 358, 375.
- Line 268. ‘working’, delete duplicated word.
- The number 2 (Line 234) should appear as a subscript. The same in Line 241
Author Response
Dear Professor:
Thank you for your comments concerning our manuscript entitled "Study on Diesel Low-Nitrogen or Nitrogen-Free Combustion Performance in Constant Volume Combustion Vessels and Contributory" (1287555). Those comments are all valuable and very helpful for revising and improving our paper, as well as the important guiding significance to our researches. We have studied comments carefully and have made correction which we hope meet with approval. Revised portion are marked in red in the manuscript. Enclosed please find the document of response to reviewer 1 comments. We sincerely hope this manuscript will be finally acceptable to be published on Atmosphere.
Thank you very much for all your help and looking forward to hearing from you soon.
Best regards!
Sincerely yours
July 10, 2021

Reviewer 2 Report
General comment:
This paper describes about the combustion performance of diesel in constant volume combustion vessels under different conditions of mixed low-nitrogen or non-nitrogen in varying proportions. In order to measure the combustion flame in the constant volume combustion chamber and analyze the effects of different compositions and concentrations in gases on combustion performance of diesel, the high-speed camera was used. The authors reported that the ignition delay period was shortened, the combustion flame was bright and slender in the condition of low nitrogen, the O2 concentration is more than 60%. The authors also reported that when the O2 concentration reached 40%, the combustion condition was optimized obviously and the combustion flame was relatively slender compared to the air working condition. With the increase of O2 concentration, the ignition delay period of nitrogen-free diesel was shortened, the duration is extended, and the combustion performance was optimized. In addition, when the O2 concentration reached 50%, with the decrease of initial temperature, the ignition delay period was prolonged and the duration was shortened obviously. When the temperature was lower than 700 K, there was no ignition. The authors concluded that the increase of diesel injection pressure was beneficial to optimize the ignition performance of diesel non-nitrogen combustion and shorten its ignition delay period and combustion duration.
In this paper, authors described the background and purpose of this paper logically based on the references in the section of introduction. The authors also explained experimental set-up and testing procedure well. And the authors drew logical conclusions based on the experimental results. Reviewer recommends this paper for journal publication with minor revisions. The authors should address the following issues.
Minor issue 1:
In Figure 5, the legend O2 21% & O2 79% should be changed to O2 21% & N2 79%.
Minor issue 2:
In general, the results of in-cylinder pressures are explained and analyzed to study the effects of experimental variations when the combustion characteristics are studied at the constant volume chamber. Please address the in-cylinder pressures after combustion with low-nitrogen or nitrogen-free conditions in the constant volume chamber. And analyze also the effects of combustion characteristics on the cylinder pressure at low-nitrogen or nitrogen-free conditions.
Minor issue 3:
The ignition delay period is a very important factor to analyze the combustion performance. There was no information how the combustion stating timing to calculate the ignition delay period define in this paper. Please address it.
Author Response
Dear Professor:
Thank you for your comments concerning our manuscript entitled "Study on Diesel Low-Nitrogen or Nitrogen-Free Combustion Performance in Constant Volume Combustion Vessels and Contributory" (1287555). Those comments are all valuable and very helpful for revising and improving our paper, as well as the important guiding significance to our researches. We have studied comments carefully and have made correction which we hope meet with approval. Revised portion are marked in red in the manuscript. Enclosed please find the document fo response to reviewer 2 comments. We sincerely hope this manuscript will be finally acceptable to be published on Atmosphere.
Thank you very much for all your help and looking forward to hearing from you soon.
Best regards!
Sincerely yours
July 12, 2021

Round 2
Reviewer 1 Report
The authors have made a considerable effort to improve the quality of their manuscript, most of the errors have been corrected, however, I consider that there is a question that has not been answered or justified yet, the comment about Figures 2 and 4, which in my opinion is relevant, has not been considered:
Any text or legend indicating the time of each image would help to understand it. In addition, a greater separation between the images corresponding to different test conditions must be done. Please modify your figures in this way.
Author Response
Dear Professor:
Thanks for your suggestions. We have adjusted Figures 2 and 4 based on your suggestions, as well as Figures 6 and 7. Please contact us if you have any questions regarding the revision of my manuscript.
Best regards!
Sincerely yours
July 14, 2021
